# OpenReview forum: "Beyond Mean Scores: Factor Models for  Reliable and Efficient AI Evaluation"
_ICLR.cc/2026/Conference — Submitted to ICLR 2026_

### Official Review · Reviewer_MSHc · 2025-10-29

**Soundness:** 1
**Presentation:** 2
**Contribution:** 1
**Rating:** 2
**Confidence:** 5

**Summary:**

This paper proposes using factor models from psychometrics to analyze AI benchmark data at the item-response level, arguing that traditional leaderboard averages obscure the multidimensional structure of model capabilities. The study demonstrates that benchmarks often conflate distinct skills and that factor-based adaptive evaluation can achieve comparable reliability with far fewer items.

**Strengths:**

1. The paper introduces a psychometric perspective to AI evaluation, moving beyond aggregate benchmark scores toward construct-level analysis.

2.  It uses large-scale, real leaderboard data from diverse benchmarks, offering empirical insight into hidden capability patterns.

3. The adaptive testing framework demonstrates substantial efficiency gains (up to 94% item reduction) while preserving measurement accuracy.

**Weaknesses:**

1. Lack of professionalism: E.g., Several figures—especially the heatmaps—lack clear color legends or scales, making them visually confusing and unprofessional. The paper dives directly into response matrices without clearly stating the evaluation objective, research questions, or intended outcomes of “AI assessment.”

2. The core approach is essentially a multidimensional Item Response Theory (IRT) model; both the factor-analytic framing and adaptive item selection are well established in prior works (e.g., [1][2]). **The methodology in this paper is almost identical to previous works, yet it neither cites them nor provides any comparative analysis**.


[1] Polo, Felipe Maia, et al. "tinyBenchmarks: evaluating LLMs with fewer examples." International Conference on Machine Learning. PMLR, 2024.

[2] Kipnis, Alex, et al. "metabench--A Sparse Benchmark to Measure General Ability in Large Language Models." arXiv preprint arXiv:2407.12844 (2024).

**Questions:**

What is difference between your factor model and IRT?

---

> ### Author Response · Authors · 2025-11-24
>
> We appreciate your thoughtful feedback. Below, we address your comments.
>
> **Weakness 1: Visualization clarity and unclear research questions.**
> Thank you for pointing this out. Our heatmaps consistently use the **`vlag` diverging colormap**
> (centered at 0: white; –1: blue; +1: red), And Figure 3 already includes a colour bar. To ensure full
> clarity, we revised and added explicit colour bars to Figure 1 and noted this convention clearly in the captions.
>
> Regarding research framing, we agree that the introduction should more clearly articulate our evaluation
> objectives. In the revision, we revised and organized the motivation. Specifically, we try to address
>
> ### **RQ1 — Can higher-rank factor models reveal multidimensional benchmark structure?**
> Prior work [1–3] uses 1-D Rasch-style IRT and assumes a single latent ability for efficient evaluation. Our focus is different:
> we ask whether higher-rank factor models can uncover multiple capability dimensions at leaderboard
> scale.
>
> ### **RQ2 — Can we get a better measurement of the construct, estimate disaggregated factor-level scores**
> After curating a validity-check dataset, we still fit a factor model with target rotation to expert-annotated data, and we observe decent alignment between our learned factor structure and prior expert-annotated cognitive constructs. (See details in general response “3. Follow-up experiment addressing concerns about spurious patterns”)
>
> ### **RQ3 —Are higher-rank factor models compatible with prior work on efficient evaluation?**
> We show that factor models remain fully compatible with efficient evaluation methods such as adaptive testing proposed in prior work [3]. By selecting items that are maximally informative for multiple latent dimensions rather than a single trait, adaptive testing enables the factor model to estimate test-taker multi-dimensional ability with far fewer items.
>
>
> —
>
> **Weakness 2 / Question 1: How does your method differ from IRT?**
> We agree with the reviewer that factor models fall within the broader IRT family. The distinction is
> in 1) *which* IRT regime we operate in and 2) our purpose using the model.
>
> Prior work focuses on **1-dimensional Rasch models** for the purpose of efficient evaluation.
>
> Our goal is to study higher-rank, multidimensional factor models, which reveal multiple capability constructs that cannot be captured under a 1-D assumption. This multidimensional analysis is central to our contributions.
>
> For a more detailed discussion comparing our work to Metabench (and Tinybench), please refer to the main response section  **2.1 Relation to IRT-style efficient  evaluation  papers**
>
> ---
>
> **References**
>
> [1] Polo et al., *tinyBenchmarks*, ICML 2024.
> [2] Kipnis et al., *metabench*, 2024.
> [3] Truong et al., *Reliable and efficient amortized model-based evaluation*, ICML 2025.

---

### Official Review · Reviewer_NFUq · 2025-10-31

**Soundness:** 3
**Presentation:** 3
**Contribution:** 3
**Rating:** 6
**Confidence:** 3

**Summary:**

This paper addresses the shortcomings of using mean scores to determine the performance of LLMs. Through factor model analysis, the authors are able to probe into different capability dimensions of the data, infer performance on held-out data-model evaluation pairs and also provide a framework for adaptive testing to reduce evaluation costs.

**Strengths:**

1. The paper does a great job with theoretical guarantees. For example, a factor model generally assumes continuous variables but this work assumes binary data which cannot be normally distributed. The way to adapt binary data to a factor model is with the tetrachoric correlation, which is detailed in the work.

2. The scale of data points and models from OpenLLM Leaderboard is large-scale which gives validity to the authors' claims.

3. Adaptive testing, while not completely novel, has been demonstrated here to provide significant reductions in evaluation costs.

**Weaknesses:**

I am very inclined to increase my score to an 8, but need the following points addressed.

1. There is no ground truth to validate that the factors extraxcted from the model correspond to useful capabilities.

2. The assumption of binary(correct/incorrect) evaluations is not  accurate: there are datasets for LLMs such as WMT that are numeric floating point scores. Large benchmark collections such as HELM contain many samples of this metric. The authors should conduct an additional study to test the factor model under heterogeneity.

3. Human experiment: there needs to be more analysis such as inter-annotator agreement and review guidelines.

4. Confounders for "models that perform well on one part of the benchmark are very likely to perform poorly on another": some confounders that can affect this claim are data contamination (training on the test set), diversity in training objectives (base model vs instruction-tuned).

5. I contest the conclusion from the MUSR example (L372-L403), specifically "two items may appear textually similar, yet test takers
can exhibit sharply different response patterns". I would argue that the factors are considering superficial reponse patterns if two very similar questions from the same narrative have different answers. I base mty argument on theory-of-mind reasoning [1]

[1] Amirizaniani et al. Can LLMs Reason Like Humans? Assessing Theory of Mind Reasoning in LLMs for Open-Ended Questions, 2024

**Questions:**

1. Real-world data is noisy and missing evalutions are likely given the rapid proliferation of models and datasets. How would the authors treat missing entries in P before the train-test split? Interpolating the value should not be a scientific solution.

2. What is the justification of error variance being fixed at $\pi^2$/3?

3. Does the promax rotation(oblique rotations) produce more interpretable factors? How does it compare to orthogonal rotations?

---

> ### Author Response · Authors · 2025-11-24
>
> We thank the reviewer for the careful and constructive feedback. Below we respond to each point in detail.
>
> ## **Weakness 1. “There is no ground truth to validate that the extracted factors correspond to useful capabilities.” & Weakness 3. There needs to be more analysis, such as inter-annotator agreement and review guidelines**
>
> We agree with the reviewer that there is no known ground truth about the true underlying capability structure of LLMs. Regarding the request for additional analysis, such as inter-annotator agreement and detailed review guidelines, we agree that these procedures are important when the goal is to establish *validated* semantic constructs. However, establishing construct validity requires a multi-step framework (content validity, convergent/discriminant validity, criterion validity). This is explicitly outside the scope of our current work.
>
> We emphasize that interpretability is not a primary contribution of this work, and we do not claim that the factors correspond to psychological or domain-grounded abilities. The small human-labeling study is included only as an exploratory illustration and is not used to support any scientific conclusion. More specifically, our primary claim on “item heterogeneity” is behaviour-driven and does not rely on interpreting the factors. Please see response to Reviewer 55PF“5. Interpretation of Latent Factors”, for a more detailed discussion.
>
> ## **Weakness 2. “Binary correctness is not accurate—LLM benchmarks contain continuous scores.”**
> We appreciate this point. However, the overwhelming majority of public LLM benchmark data is binary (correct/incorrect). Existing continuous-scored benchmarks tend to have far fewer items or models, which makes fitting multidimensional factor models statistically unreliable and likely underpowered. Meanwhile, the core finding of the paper—the presence of heterogeneous item structures that cannot be captured mean score—is a property of the item space, not of the binary scoring format. Heterogeneity arises from variation in item capability demands and test-taker capabilities, and this phenomenon should persist regardless of whether items are scored as binary, partial credit, or continuous performance metrics.
>
> We agree that evaluating graded or continuous scoring would be a valuable next step, and we view it as a promising direction for future work once larger continuous-scoring datasets become available. It would also be particularly interesting to test whether the item heterogeneity we identify generalizes to datasets whose outcomes are not binary, which we expect should hold given that heterogeneity reflects underlying capability structure and behaviour pattern rather than the specific scoring scale.
>
> ## **Weakness 4. Confounders for negative correlations (contamination, specialization, training objectives)**
> The reviewer suggests that negative correlations might arise from:
> - Data contamination,
> - Divergent training objectives (base vs. SFT vs. RLHF).
>
> We appreciate the opportunity to clarify why these confounders are unlikely to explain our observations.
> 1. **Contamination → negative correlations?**
> In our data, negative correlations arise even between nearly identical items (e.g., MUSR’s “Mr Brown” and “Emma” variants). These items have extremely similar text and come from the *same narrative*, making partial contamination an implausible explanation.
>
> 2. **Training specialization → negative correlations?**
> One mechanism by which differences in training or post-training could induce negative correlations is if the models were differently specialized.  For example, suppose that one model was trained to be good at biology but poor at history, and another was trained to be good at history but poor at biology.  Then our modeling could surface negative correlations in model performance between a “biology” factor and a “history” factor.  However, the negative correlations we observe occur within sets of items that are nearly semantically identical. Differences in training specialization cannot explain the observed negative relations.

---

> ### Author Response · Authors · 2025-11-24
>
> ## **Weakness 5. Regarding the MUSR example**
> We appreciate the reviewer’s thoughtful objection. We agree that, in this specific case, the factor model may be capturing superficial behavioural patterns rather than deep Theory-of-Mind reasoning. Two items that are textually similar but yield different responses can indeed reflect method variance rather than construct variance.
>
> At the same time, the broader heterogeneity results do not rely on the MUSR narrative example. After performing additional data cleaning to reduce validity and address the superficial pattern concern—merging variants of questions via conjunctive scoring, and ensuring test-taker diversity—we still observe strong evidence of **item-level heterogeneity** across multiple benchmarks and domains. This supports our main claim about benchmark items heterogeneity, independent of whether the MUSR example reflects genuine ToM constructs. For the details of this study, please see the general response section **“3. Follow-up experiment addressing concerns about spurious patterns”**.
>
>
> In short, while the MUSR example may exhibit superficial patterns and we acknowledge its limitations, the core heterogeneity findings remain robust and are demonstrated through additional analyses on curated and validity-checked data.
>
> ## **Q1. Missing data before train/test split**
> ### “Interpolation should not be a scientific solution.”
>
> We agree, and we **do not interpolate** correctness. Our handling of missing entries:
> - Missing values are stored as `NaN`,
> - The likelihood and gradients are masked so that only observed entries contribute,
> - No imputation is performed.
> - They are not part of the calculation during evaluation, say computing AUC on test set and are not part of training
>
> ## **Q2. What is the justification for the error variance being fixed at  π²/3?**
>
>
> We model the latent error term using a logistic distribution with location $0$
> and scale $s$, i.e., $\mathrm{Logistic}(0, s)$. A logistic distribution with
> scale $s$ has variance $s^2 \pi^2 / 3$.
> The standard logistic function corresponds to the unit-scale logistic
> distribution, where $s = 1$. Substituting $s = 1$ yields the fixed variance
> $$
> \mathrm{Var}(\epsilon) = \frac{\pi^2}{3}.
> $$
>
>
>
>
>
> ## **Q3. “Does promax (oblique) rotation improve interpretability compared to orthogonal rotations?”**
> Yes.  We adopt **promax/quartimin** for the specific factors because:
> - Oblique rotation allows correlated factors,
> - Capability dimensions typically co-occur (e.g., knowledge of natural science + knowledge of applied science),
> - Orthogonal rotations unrealistically force independence.

---

### Official Review · Reviewer_wmHA · 2025-10-31

**Soundness:** 3
**Presentation:** 3
**Contribution:** 2
**Rating:** 2
**Confidence:** 3

**Summary:**

This paper applies factor models to analyze 4,416 language models on 21,176 questions from six benchmarks. The work demonstrates that: (1) benchmarks contain distinct, sometimes negatively correlated constructs that mean scores conflate; (2) factor models enable adaptive testing that reduces evaluation costs, and (3) factors can be interpreted via clustering and LLM-based labeling. The paper positions this as bringing psychometric methodology to leaderboard-scale AI evaluation.

**Strengths:**

**S1: Strong empirical validation and scale.** The analysis of 4,416×21,176 item responses seems to be substantially larger than prior work. The factor models achieve good predictive performance.

**S2: Compelling practical impact.** The adaptive testing results demonstrate 33.9-94.3% reduction in required questions while maintaining 95% reliability.

**S3: Clear presentation.** The paper is well-written (even though one could probably reduce the equation-load without affecting understandability) with effective visualizations (especially Figure 1).

**Weaknesses:**

### W1: Overclaimed Novelty

Line 104 claims: "We present a first systematic attempt to interpret factors discovered by the model."

This is might not be completely true given existing work:
- [**CAIMIRA (Gor et al.)**](https://arxiv.org/abs/2410.06524) already provides systematic interpretation of QA abilities using IRT with theory-driven constructs linked to cognitive science frameworks
- **metabench** factor analysis (arxiv.org/abs/2407.12844v2) applies factor analysis to benchmark interpretation
- Additional relevant work: arxiv.org/pdf/2501.17200, sciencedirect.com/science/article/pii/S0160289624000527

These papers are not cited. The interpretation contribution should be reframed as "applying interpretation at leaderboard scale" rather than claiming primacy.
No comparisons to those approaches are made, and only a mean baseline is used.

The motivation that mean scores are inadequate  is well-established (cited: Saxon et al. 2024, see also https://www.sciencedirect.com/science/article/pii/S0927025625003842); the contribution seems to be applying factor models at scale, not identifying this problem


### W2: Weak Interpretation Methodology

The interpretation approach has no evidence that labels are stable, expert-validated, or predictive beyond the clusters they describe

This is notably weaker than theory-driven approaches in prior work like CAIMIRA.

**Questions:**

1. How does your work contextualize with CAIMIRA, metabench, and co?
2. Have you validated GPT-5 labels in some way, e.g., with domain experts?
3. How do you ensure factors measure constructs rather than spurious artifacts (given the MUSR pattern-matching evidence)?

---

> ### Author Response · Authors · 2025-11-24
>
> We thank the reviewer for the thoughtful and constructive feedback. We address each point below.
>
> ---
> ### **W1: Overclaimed Novelty & Q1: Relationship to CAIMIRA, Metabench, and related work**
>
> We appreciate the reviewer highlighting CAIMIRA, Metabench, and other recent applications of factor analysis and multidimensional IRT. We fully agree that there has been prior work exploring interpreting latent dimensions.
>
> **Clarification:**
> Compared to other papers that attempt to provide an interpretation of benchmarks, our approach is behaviour-driven instead of semantics-driven. This leads to a resulting structure that reflects **behavioural regularities** rather than semantic or topical similarity. This provides a behavioural perspective that is complementary to semantic methods: textually different items may elicit similar model behaviour, and sometimes, textually similar items may diverge sharply in practice. A behaviour-only view does not naturally lead to semantic interpretability in the way that prior semantic-based methods do. However, it does reveal behavioural regularities that semantic methods may overlook.
>
> We revised and made this tradeoff explicit and emphasized that we offer only an initial attempt to understand the factors—fully linking behavioural dimensions to semantic constructs would require a full validity-theory framework and is beyond the scope of this project.
>
> Below, I provide a comparison with the prior works that the reviewer mentioned.
>
> - **CAIMIRA**: The paper decided to move away from the multidimensional IRT framework because of the challenges in identifiability, and therefore introduced a redesigned framework. We apply proper oblique rotation to address identification and enhance interpretation. In addition, the CAIMIRA framework can still be confounded. Because the model is still trained via a maximum-likelihood objective, the parameters must absorb any systematic variance in the response patterns. As a result, confounders and superficial behavioural patterns can be captured as latent dimensions. Downstream, these confounded directions may then be misinterpreted as meaningful semantic factors, even though they arise from artifacts rather than underlying capabilities.
>
> - **Metabench**: The paper applies factor analysis on coarse-grained benchmark scores and focuses on compressing benchmarks and uncovering redundant components. Please see the full comparison in the general response section **2.1 Relation to IRT-style efficient evaluation papers**
>
> Our goal is not to interpret latent traits and enable efficient evaluation but to show that aggregating heterogeneous items into a single mean score is problematic.
>
> - **Lessons from the trenches on evaluating machine learning systems in materials science**: This paper discusses the idea that aggregation or mean score is inadequate. More specifically, it makes two arguments: i) measurement is a dimensionality reduction. ii) Averaging score across tasks with different performance ceilings may lead to artificially overweighing of tasks with a larger range.
>
> Their first argument is that measurement is a dimensionality reduction. While true, this statement applies to any measurement metric or evaluation method, as they, by definition, are a dimension reduction of the test-taker-item response. This is conceptually correct but too general to be actionable: it does not identify which aspects of the dimensionality reduction create problems, how large those problems are, or how one should model the underlying structure that has been lost.
>
> Their second argument is a legitimate concern and is a known motivation for adopting Rasch-style IRT. A 1D Rasch model corrects for task-difficulty imbalance by estimating an item difficulty parameter and giving test-takers an ability parameter on a single latent dimension. But crucially, the Rasch correction assumes all items measure the same underlying trait. The argument concerns differences in scale and difficulty, not multidimensionality.
>
>
>
>
> We updated our related work section to include:
> - CAIMIRA (Gor et al. 2024)
> - Metabench (Kipnis et al. 2024)
> - Lessons from the trenches on evaluating machine learning systems in materials science (Alampara, Schilling-Wilhelmi, & Jablonka, 2025)
> and we clarify the distinctions in scope.

---

> ### Author Response · Authors · 2025-11-24
>
> ### **W2: Weak Interpretation Methodology & Q2: Validation of GPT-5 labels**
>
> We agree with the reviewer that our labelling procedure is not theory-driven and is not intended to establish latent constructs in a psychometric sense. As mentioned in the previous section, a behaviour-driven approach does not naturally lead to semantic interpretability in the way that prior semantic-based methods do.
>
> Please see more discussions in response to Reviewer 55PF “5. Interpretation of Latent Factors”, GPT-5-generated labels are used only for exploratory purposes.  However, we do not assert that the discovered dimensions correspond to semantic or cognitive constructs, no conclusions in this paper about “item heterogeneity” rely on validated semantic interpretation, and connecting behavioural factors to validated cognitive constructs would require a broader validity-theory framework, which is beyond the scope of this work.
>
> To avoid misunderstanding, we revised the interpretation section to:
> - Clearly state that our labelling is exploratory and non-essential,
> - Emphasize that our main technical contribution is discovering item heterogeneity and not factor interpretation.
>
> ---
>
> ### **Q3: Ensuring factors capture constructs rather than spurious artifacts**
>
> In some cases, the factor model may be capturing superficial behavioural patterns. We have conducted additional experiments to illustrate item heterogeneity while addressing spurious artifact concerns.
>
> Here is a brief summary:
> 1. We first curated a dataset designed to mitigate spurious pattern concerns by using conjunctive scoring on groups of questions that are formatting perturbations of each other. This helps remove spurious correlations between items.
> 2. We then conducted parallel analysis, which indicates that the curated dataset exhibits multiple factors.
> 3. Next, we perform an EFA  and compare our learned specific factors with the human-annotated dimensions from a paper in which psychometricians annotated key constructs on MMLU-Pro. The specific factors from our model aligned well with these expert-defined dimensions. This provides additional evidence that our fitted factors recover meaningful latent skills rather than artifacts of surface-level item structure.
>
> More details can be seen in the main response summary section **3. Follow-up experiment addressing concerns about spurious patterns**
>
> ---
>
> We thank the reviewer again for raising these important points.
> We revised the manuscript to:
> - Contextualize the work with prior works in efficient evaluation and interpretation
> - Clarify the exploratory nature of any interpretation labeling,
> - include the new analysis showing item heterogeneity while addressing the spurious pattern concern

---

### Official Review · Reviewer_55PF · 2025-11-02

**Soundness:** 1
**Presentation:** 2
**Contribution:** 2
**Rating:** 2
**Confidence:** 4

**Summary:**

This work applies latent factor models to LLM benchmarking. The main claimed contributions are: (1) identifying that benchmarks are testing multiple underlying capabilities that can be discovered through fitting factor models, an argument against standard mean aggregation which hides tradeoffs between these capabilities, (2) showing how factor models can be used for efficient adaptive evaluation, with the main augmenting showing one can preserve evaluation reliability while cutting evaluation costs, and (3) a case study interpreting the latent factors on LLM benchmarks using clustering based on learned factors and prompting GPT 5 on clusters.

Their experiment involves taking 6 LLM benchmarks (e.g. MMLU Pro, BBH, IFeval, etc) and many LLM responses on those benchmarks sourced from the open LLM leaderboard. They fit a multidimensional factor model to each benchmark dataset, holding out some instances for testing.

**Strengths:**

This work is a tutorial-like deep dive of factor analysis and how to use it on LM benchmarks to gain different insights. It covers a breadth of topics and analyses. Factor analysis methodology is sound.

**Weaknesses:**

**Lacking proper dev set for tuning rank of factor models**
Something concerning to me is that their factor models lack a proper baseline for comparison. In L269, they describe how they split the data into train 80% and test 20% for each benchmark, and fit factor models that result in high AUC (mid-high 90%s) across benchmarks. The problem is they also tune the hyperparameter (rank size) of their factor models on this test set (Appendix C.1) rather than splitting off a proper development set from the train set. So these AUC scores are going to be artificially inflated.

**Lacking baseline for factor models**
The only baseline reported in this work for proving the validity of the factor models is a super naive baseline that takes the mean performance of the chosen subset to predict the test instances. Indeed this is the “standard” practice in LM benchmarking today, but there are many works that already improve on this. For example, why not have results against Truong et al (2025)’s use of Rasch style models? If this work is arguing that single-trait assumption is limiting (L135), and multi latent factor models are superior, then one should show it empirically by comparing simple Rasch model or 2PL models in the same experimental setup and showing it has lower AUC than the factor model. It is the responsibility of the authors to pick reasonable baselines that can help readers understand the strengths/weaknesses of the proposed method.

**Lacking baselines for analysis/insights on decomposed factors**
One of the big results of this paper is in Figure 3, showing how factor models can be used to produce interesting clusters with good block-diagonal membership diagnostics. The issue here is, again, there are lacking baselines that show that factor models are needed to produce these types of insights. Can one have achieved this with PCA? With some more naive embedding + clustering technique? Are there metadata tags on instances in these benchmarks that could’ve done better? It’s nice that factor models can reveal this type of structure, but one needs to show that this structure is non-trivial to recover using reasonable other methods.  Especially if this experiment is achieved with a rank 2 factor model; would this be achievable with the first two components of PCA, or even clustering on the two learned variables in a 2PL model?

Furthermore, on the topic of decomposing benchmark tested capabilities, there is prior work such as “FLASK: Fine-grained Language Model Evaluation based on Alignment Skill Sets (ICLR 2023)” that exactly tackled this problem of decomposing aggregate scores into finer-grained skills. Another work is “Unearthing Skill-Level Insights for Understanding Trade-Offs of Foundation Models (ICLR 2024)” that also performed this type of decomposition to discover subsets of benchmarks that test finer grained skills in LM evaluation. Indeed, they do not apply factor analysis, so this work is novel in that regard, but because there is prior work in this space, it is important for the submission authors to situation this work against those prior works. Missing also from related work is “Anchor Points: Benchmarking Models with Much Fewer Examples (Vivek et al 2024)” which also uses item clustering.

**Lacking baselines for efficient LM evaluation**
This work shows an adaptive testing procedure for efficient LM evaluation (Sec 4.2). This is an incredibly short section, but again, it lacks baselines and situating in prior literature. There are many works on efficient benchmarking, some even using IRT which is related inspiration from psychometrics. There is no citation in this work for Anchor Points (Vivek et al 2024), Tiny Benchmarks (Polo et al 2024), Metabench (Kipnis et al 2025), and more. All of these build on ideas of selecting an informative subset of items because they reflect some “underlying” capabilities. The factor analysis approach is new, but there is so much work uncited & not baselined against in this paper.

**Lacking related work and baselines for interpretable features**
With any sort of work that’s about using learned latent factors and assigning human interpretable meaning to these learned factors, there is already plenty of work in this space that is not cited or baseline against. This paper states “We view this as an initial
step toward a richer interpretive framework that future work can refine and extend” in L433. Unfortunately, this work is not the initial step at all.  “Qualeval: Qualitative evaluation for model improvement (Murahari et al 2024)” and “Unearthing skill-level in-
sights for understanding trade-offs of foundation models (Moayeri et al 2024)” automatically categorize benchmark instances into capability groups. “Goal-driven explainable clustering via language descriptions (Wang et al 2023) and “ Explaining datasets in words:
Statistical models with natural language parameters (Zhong et al 2024)” are also examples. There should be better related work discussion + baselining against them.

**Questions:**

Sec 3 takes up a ton of space mostly explaining background on how factor analysis works. It’s not clear this is needed, and if anything could be shortened to make room for more important additions to this work, especially baseline experiments.

---

> ### Author Response · Authors · 2025-11-24
>
> We thank the reviewer for the extensive and thoughtful feedback. We address each concern in detail below, and appreciate the opportunity to clarify our contributions and add missing baselines and context.
>
> ---
>
> ## **1. Rank Selection and Use of Validation Set**
>
> We agree that hyperparameter tuning on the test set is not appropriate for evaluating predictive performance. But to clarify, the goal of the rank-sweep experiment is **not** to optimize test AUC, but to illustrate that **different ranks induce different latent structures**—a central point of the paper. Because the experiment’s purpose is to explore model properties across ranks—and *not* to perform model selection— a separate development split is not required.
>
> ---
>
> ## **2. Missing Baselines for Factor Modeling**
> We appreciate this suggestion and have added:
> **Rasch Baseline**
> As expected, the classical Rasch models perform worse than the factor model in AUC and item-level predictive error, supporting the claim that single-trait models are insufficient for heterogeneous benchmarks. We thank the reviewer for motivating this addition. In this experiment, we train a Rasch model under the *Random Mask* scheme, where 20% of responses are held out as a test set, and we compare the resulting test-set AUC against higher-rank factor models (Rasch model performance is indicated with the x-axis label “Ra”).
> [Figure 1 (OSF)](https://osf.io/dnr6q/files/vfqab?view_only=3e08196565ad49e59b7e02b123f19b4b)
>
> ---
> ## **3. Baselines for Factor-Based Clustering**
> We have added “PCA” baselines:
> We project items using the first two principal components and cluster the projections using GMM. To evaluate cluster stability, we perform two 90% subsamplings and compute the cluster-overlap matrix[Figure 2]:  each row corresponds to one cluster in subsample A, and each entry reports the percentage of items in that cluster that also fall into each cluster of subsample B.
>
> If PCA captured a stable underlying structure, this overlap matrix would be close to an identity matrix—each cluster in subsample A would correspond cleanly to exactly one cluster in subsample B. However, as shown in Figure 2, the PCA clusters are unstable: the overlap matrix is noisy, with individual clusters frequently splitting across multiple clusters in the other subsample.
>
> This instability suggests that PCA fails to recover the coherent, stable structure revealed by our factor–model–based clustering. [Figure 2 (OSF)](https://osf.io/dnr6q/files/t7j2e?view_only=3e08196565ad49e59b7e02b123f19b4b)
>
> ## **4. Baselines and Related Work for Adaptive Testing**
>
> Regarding efficient LM evaluation, our goal is not to introduce a new state-of-the-art method for speeding up evaluation. Existing approaches—such as Anchor Points, TinyBench, and MetaBench—are specifically designed for evaluation efficiency, whereas our work focuses on item heterogeneity and the multidimensional structure. Our intention in this section is simply to illustrate that our higher-rank factor model remains compatible with adaptive testing frameworks such as CAT. See details in general response “2.1 Relation to IRT-style efficient  evaluation  papers”

---

> > ### Author Response · Authors · 2025-11-24
> >
> > ## **5. Interpretation of Latent Factors**
> >
> > Our paper does not claim that factor loadings correspond to cognitive or
> > semantic constructs, nor do we claim validated interpretability. The reviewer is correct
> > that prior work [1-4] has developed frameworks for interpreting latent structure.
> >
> > At the same time, we would like to clarify that our approach differs fundamentally from these prior methods.
> > The cited papers follow a *semantic-first* pipeline: they begin from item text, LLM-derived embeddings, or other semantic features, and then perform clustering or categorization based on those representations.
> >
> > In contrast, our factors are derived purely from behavioural response patterns—the joint structure of model correctness across items—without using item text or metadata. This leads to several qualitative differences:
> >
> > - Semantically different items can cluster together if models exhibit similar behavioural difficulty or error patterns, as they might test similar latent ability
> > - Semantically similar items may fall into different clusters if models respond to them differently.
> > - The resulting structure reflects **behavioural regularities** rather than semantic or topical similarity.
> >
> > We acknowledge that there has been an attempt to cluster and interpret benchmarks based on semantics, but we are making initial steps to interpret the discovered behaviour-driven factors. Because of the approach being behaviour-driven, our approach does not naturally yield semantic interpretability in the same way as semantic-based methods do. We therefore do not claim “a definite optimal interpretation”.  Importantly, no conclusions in this paper about “item heterogeneity” rely on validated semantic interpretations.
> >
> > To prevent misunderstanding, we revised the manuscript to:
> >
> > - Remove any implication that interpretability is a primary contribution,
> > - Explicitly distinguish behavioural-factor analysis from semantic clustering approaches,  and contextualize previous studies
> > - Clarify that connecting behavioural factors to validated cognitive constructs would require a broader validity-theory framework, which is beyond the scope of this work.
> >
> > ---
> >
> > ## **6. Regarding Section 3 Length**
> >
> > We appreciate this feedback and shortened Section 3. Discussions like deriving the structure matrix and tetrachoric correlation can be shortened and moved to the appendix. This makes room for new baseline experiments discussed in general response and expanded discussions of related work.
> >
> > ---
> >
> > We thank the reviewer again for the detailed comments. We have substantially expanded the baseline experiments (Rasch, PCA), added missing related work, clarified the scope of our contributions, and contextualized them with prior works.
> >
> >
> > [1] Murahari et al., *QualEval: Qualitative Evaluation for Model Improvement*, 2024.
> > [2] Moayeri et al., *Unearthing Skill-Level Insights for Understanding Trade-Offs of Foundation Models*, 2024.
> > [3] Wang et al., *Goal-Driven Explainable Clustering via Language Descriptions*, 2023.
> > [4] Zhong et al., *Explaining Datasets in Words: Statistical Models with Natural Language Parameters*, 2024.

---

### Author Response · Authors · 2025-11-24
**General Response**

We thank the reviewers for their thoughtful and constructive feedback. Below, we summarize the main points of clarification and describe how we have strengthened the paper in response.

## **1. Clarifying the main contribution**

Our paper’s primary contribution is to show that benchmark evaluation should *go beyond mean scores* (which includes one-dimensional Rasch models), because of item heterogeneity. Our contribution is not interpretability or efficiency per se.

More concretely,

- We advocate that a disaggregated score is necessary because of item heterogeneity.
- The disaggregated scores can be measured by a factor model, which is behaviour-driven and is fitted to a dataset that has done proper deconfounding. The factors have been found to have decent agreement with expert-driven annotation. (details in general response “3. Follow-up experiment addressing concerns about spurious patterns”).
- The factor model remains compatible with adaptive testing to estimate disaggregated factor-level scores.

---

## **2. Contextualizing our approach relative to prior work**

### **2.1 Relation to IRT-style efficient  evaluation  papers**
We recognize the reviewers’ very valid concern that we did not sufficiently contextualize our contribution within related work.  As noted by 55PF,  NFUq, and MSHc, the metabench paper appears to be especially closely related to our work. Metabench relies on a similar dataset: they use the non-archived Open LLM Leaderboard data for their analysis, whereas we use the archived one.  These two leaderboards have some but not all benchmarks in common.  The authors show that IRT methods can be used to estimate model performance with only ~100-200 strategically selected sub-items, and it is unnecessary to run the full benchmark.  The ultimate goal of metabench (and tinyBenchmarks), however, is to *accurately estimate the overall benchmark score–what we term the “mean score”-with as few items as possible*.  As we indicate in the title, the fundamental goal of our paper is to *go beyond mean scores.*  Whereas metabench shows that one can reduce RMSE in estimating a model’s overall score on MMLU-Pro from 1.908 under random item subsampling to 1.743 with a 1-d IRT model on the just that benchmark and 1.430 by jointly modeling all benchmarks, we argue that *it is reductive to report a single mean score for MMLU Pro because further inspection shows that the data are not consistent with a single-trait 1-d capability model.*  MMLU-Pro, like MUSR, can be broken down into subgroups of items on which model responses are significantly *negatively correlated* in their responses.  If all we care about is the aggregate benchmark mean score, it doesn’t matter if we use a 1-d model or a higher-dimensional model.  But our paper argues that we shouldn’t just be caring about mean scores in the first place.

As noted in our response to Reviewer 55PF “4. Baselines and Related Work for Adaptive Testing”, we demonstrate that our factor model enables efficient adaptive sampling to estimate a multi-dimensional measure of model performance.  That is, the techniques used in metabench and tinyBenchmarks to efficiently estimate *mean scores* using a 1-d model can also be applied to estimate *disaggregated factor-level scores*


### **2.2 Relation to semantic-based interpretability methods**
The interpretability papers cited by reviewers [1–5] generally begin with text, text embeddings, or LLM-extracted features, and then cluster or derive semantic groupings.
Our approach is fundamentally different:
We derive factors purely from model response patterns, independent of item text, providing a behavioural perspective rather than a semantic one.

This means:
- semantically similar items may have models respond to them differently
- semantically different items may cluster together due to shared behavioural patterns.

A behaviour-only approach does not yield semantic interpretability in the same way as text-based methods, and we acknowledge this tradeoff. However, it captures behavioural regularities that semantic methods may not detect. We emphasize that our goal is not to provide final semantic interpretations of factors, but to establish the existence and stability of heterogeneous behavioural dimensions across models. This evidence does not rely on validated semantic interpretations.

---

> ### Author Response · Authors · 2025-11-24
> **General Response (Part 2 of 3)**
>
> ### **2.3 Construct variance vs. method variance**
> Reviewers raised the important question of whether our factors capture “real” constructs or superficial patterns.  In psychometrics, item responses reflect both:
>
> - **construct variance** — ability we aim to measure
> - **method variance** — systematic but construct-irrelevant patterns
>
> LLMs exhibit analogous method variance (format artifacts, stylistic patterns, CoT defaults, etc.). Meanwhile:
>
> - **(a)** some method variance should be mitigated (e.g., using conjunctive scoring to merge augmentation-derived items), but
> - **(b)** some method variance is predictively useful and should not always be removed.
>
> This also explains why behaviour-derived clusters may be less semantically interpretable, while still being valuable and necessary for capturing the true structure of model capabilities. Fully mapping behavioural factors to semantic constructs would require a full validity-theoretic program, which is beyond our scope.
>
> ## **3. Follow-up experiment addressing concerns about spurious patterns**
>
> Reviewer wmHA and reviewer NFUq suggested that some factors may be driven by spurious patterns. To address this, we conducted a follow-up experiment,
> This experiment:
>
> 1. Reduces spurious patterns by grouping all formatting-perturbation variants via conjunctive scoring, ensuring that augmentation artifacts do not create spurious factors.
> 2. Applies parallel analysis, which still identifies multiple retained factors, indicating persistent multidimensional structure even under reduced noise.
> 3. performs an EFA and compares our behaviour-derived factors with the human-annotated skill categories from [6]. We observe a **strong correspondence** between behaviour-driven dimensions and expert-labelled categories.
>
> This follow-up study directly addresses concerns about superficial patterns. Even after aggressively reducing superficial patterns, we still find clear evidence of multidimensionality and heterogeneous capability structure.

---

> ### Author Response · Authors · 2025-11-24
> **General Response (Part 3 of 3)**
>
> ### Curating Dataset with Valid Items and Diverse Test Takers
> #### 1. Identifying items that test the same construct
> We first clustered items using Gemini embeddings and cosine similarity.  Items with cosine similarity > 0.99 were found to be the same questions’ perturbed versions and semantically tested the same skill.
>
> #### 2. Conjunctive scoring for perturbed items
> For a question with multiple perturbed versions, we aggregated responses using conjunctive scoring:
>
> - For each test taker, all neighbouring perturbed items were collected.
> - An AND rule was applied:
>   If *any* perturbed version of the question was answered incorrectly, the test taker was marked as incorrect on the underlying question.
>
> Conjunctive (AND) rules are widely studied in the psychometrics literature[6]. They are known to reduce false positives from chance performance. Our application of the AND rule to perturbed versions of the same item follows this principle: if a model fails any perturbation, it is evidence of instability on the underlying construct.
>
> At this point, we have curated a subset of items—S1—for which conjunctive scoring has been applied to aggregate perturbed versions into a single underlying item. All remaining items form the complementary set S2. We will refer to these two subsets as S1 and S2 throughout the remainder of the discussion.
>
> #### 3. Ensuring test-taker diversity
> We observed a large block of ~1,000 finetuned models (mostly Qwen 2.5 14B variants) with extremely high similarity in response patterns. To prevent overrepresentation of near-duplicate test takers, we removed any model whose tetrachoric correlation with another exceeded 0.9.
>
>
> ---
>
> ### Discovering the Number of Latent Factors
>
> After curating a dataset with valid items and diverse test takers, we performed parallel analysis to estimate the intrinsic dimensionality of the response matrix.
>
> - Both real and simulated datasets were 80% subsampled 1,000 times.
> - For each factor, we compared the 99% confidence interval (CI) of the real eigenvalues with the 99% CI of the simulated eigenvalues.
> - A factor was retained if the lower bound of the real-data CI exceeded the upper bound of the simulated-data CI.
>
> The scree plots indicate the presence of up to 21 meaningful factors, providing strong evidence for a multi-factor structure rather than a purely unidimensional construct.  See the scree plot here: [Scree Plot (OSF)](https://osf.io/dnr6q/files/tjgvf?view_only=3e08196565ad49e59b7e02b123f19b4b)
>
> ---
>
> ### Performing an efa and aligning with Prior Human Annotations
>
> A factor model with 18 dimensions is fitted to the curated dataset on S1 (using a random mask where 20% entries are masked out as test set) and achieved auc of 0.86 Freezing the fitted test taker ability parameter U, we can fit to the response matrix of the complementary dataset S2 to get item factor loading on S2 items. The fitting resulting AUC is 0.88, which indicates the factor was able to generalize.
>
> After fitting factor loading to the S1 and S2 datasets, we evaluated semantic interpretability by aligning the learned factors to the expert-annotated 18 dimensions proposed in Zhou et al. (2025) [7] in the MMLU-PRO dataset. We first performed target rotation on 80% of the factor loading matrix (training split) to match the human-annotated dimension. This yielded a 0.3 correlation on the training split. Applying the learned rotation to the held-out 20% produced a similar 0.28 correlation, indicating stable alignment.
> These results show that the learned factors achieve meaningful semantic alignment with human-annotated dimensions, even under different modelling methods.
>
> ---
>
> We hope this summary clarifies our contributions and addresses the reviewers’ questions thoughtfully. We have incorporated these improvements into the revised paper.
>
> **References:**
> [1] Murahari et al., *QualEval: Qualitative Evaluation for Model Improvement*, 2024.
> [2] Moayeri et al., *Unearthing Skill-Level Insights for Understanding Trade-Offs of Foundation Models*, 2024.
> [3] Wang et al., *Goal-Driven Explainable Clustering via Language Descriptions*, 2023.
> [4] Zhong et al., *Explaining Datasets in Words: Statistical Models with Natural Language Parameters*, 2024.
> [5] Gor et al., *Do great minds think alike? Investigating Human-AI Complementarity in Question Answering with CAIMIRA*, 2023.
>
> [6] McBee et al., *Combining Scores in Multiple-Criteria Assessment Systems: The Impact of Combination Rule*, Gifted Child Quarterly, 2014.
>
> [7] Zhou et al., *General Scales Unlock AI Evaluation with Explanatory and Predictive Power*, arXiv 2025.

---

### Meta-Review · Area_Chair_5cEL · 2026-01-07

**Summary:**

Paper argues mean benchmark scores hide multidimensional capability structure, and fits leaderboard-scale factor models (4,416 models × 21,176 items) to (i) expose latent (sometimes negatively correlated) constructs and (ii) enable adaptive evaluation to cut cost while preserving rankings. Reviewers split: two rejects (55PF, wmHA), one strong reject (MSHc), one weak accept (NFUq=6). Main friction is novelty/positioning vs prior IRT/metabench/tinyBench/CAIMIRA, plus evaluation hygiene and interpretability/validity claims.

**Reviewer Concerns:**

What rebuttal addressed:

Added missing related work context (metabench/tinyBench/CAIMIRA etc.) and reframed contribution as “beyond mean scores via item heterogeneity”, not interpretability per se.

Added key baselines: Rasch vs factor models (AUC) and PCA clustering stability baseline; shortened background section.

Added a spurious-pattern mitigation follow-up (conjunctive scoring over perturbations, remove near-duplicate models), parallel analysis showing multi-factor structure persists, and some alignment with expert-annotated dimensions.

Remaining concerns:

Methodology rigor: one reviewer still flags tuning/selection practices and asks for cleaner model selection (dev split) / clearer experimental protocol; rebuttal partially disputes necessity but risk remains.

Construct validity: factors may still reflect “method variance” (format/artifacts, contamination/specialization), and interpretation via GPT-5 is explicitly exploratory; some reviewers wanted stronger validation.

Novelty: substantial overlap with multidimensional IRT framing; contribution may read incremental without a sharper “what is new beyond existing IRT/benchmark-compression works” narrative.

**Reviewer Scores:**

Score-change guess (post-rebuttal):

55PF: 2 → 3 (baseline additions help, but core rigor/positioning concerns likely remain)

wmHA: 2 → 3 (novelty reframing + spurious-pattern work helps a bit)

MSHc: 2 → 2 (very certain; may not move much)

NFUq: 6 → 7 (many of their questions answered; likely bumps up)

---

### Decision · Program_Chairs · 2026-01-26

Reject